# MiR193a Modulation and Podocyte Phenotype [note 1]

**DOI:** 10.3390/cells9041004

**Published:** 2020-04-17

**Authors:** Alok Jha, Shourav Saha, Kamesh Ayasolla, Himanshu Vashistha, Ashwani Malhotra, Karl Skorecki, Pravin C. Singhal

**Affiliations:** 1Institute of Molecular Medicine, Feinstein Institute for Medical Research and Zucker School of Medicine at Hofstra-North well, New York, NY 11030, USA; ajha1@northwell.edu (A.J.); ssaha1@student.gn.k12.ny.us (S.S.); kayyasola@northwell.edu (K.A.); hvashisth@northwell.edu (H.V.); amalhotra1@northwell.edu (A.M.); 2Technion – Israel Institute of Technology, and Rambam Health Care Campus, Haifa 2710000, Israel; karl.skorecki@biu.ac.il

**Keywords:** miR193a, APOL1, Sox2, YY1, WT1, VDR, RXR

## Abstract

Apolipoprotein L1 (APOL1)-miR193a axis has been reported to play a role in the maintenance of podocyte homeostasis. In the present study, we analyzed transcription factors relevant to miR193a in human podocytes and their effects on podocytes’ molecular phenotype. The motif scan of the miR193a gene provided information about transcription factors, including YY1, WT1, Sox2, and VDR-RXR heterodimer, which could potentially bind to the miR193a promoter region to regulate miR193a expression. All structure models of these transcription factors and the tertiary structures of the miR193a promoter region were generated and refined using computational tools. The DNA-protein complexes of the miR193a promoter region and transcription factors were created using a docking approach. To determine the modulatory role of miR193a on *APOL1* mRNA, the structural components of APOL1 3’ UTR and miR193a-5p were studied. Molecular Dynamic (MD) simulations validated interactions between miR193a and YY1/WT1/Sox2/VDR/APOL1 3′ UTR region. Undifferentiated podocytes (UPDs) displayed enhanced miR193a, YY1, and Sox2 but attenuated WT1, VDR, and APOL1 expressions, whereas differentiated podocytes (DPDs) exhibited attenuated miR193a, YY1, and Sox2 but increased WT1, VDR, APOL1 expressions. Inhibition of miR193a in UPDs enhanced the expression of APOL1 as well as of podocyte molecular markers; on the other hand, DPD-transfected with miR193a plasmid showed downing of *APOL1* as well as podocyte molecular markers suggesting a causal relationship between miR193a and podocyte molecular markers. Silencing of YY1 and Sox2 in UPDs decreased the expression of miR193a but increased the expression of VDR, and CD2AP (a marker of DPDs); in contrast, silencing of WT1 and VDR in DPDs enhanced the expression of miR193a, YY1, and Sox2. Since miR193a-downing by Vitamin D receptor (VDR) agonist not only enhanced the mRNA expression of *APOL1* but also of podocyte differentiating markers, suggest that down-regulation of miR193a could be used to enhance the expression of podocyte differentiating markers as a therapeutic strategy.

## 1. Introduction

MicroRNAs (miRNAs) are short (18–25 nucleotides, nt) single-stranded non-coding RNAs that regulate gene expression post-transcriptionally, usually through binding to the 3’-UTR (untranslated region) of their target mRNAs. They repress protein production by destabilizing the messenger RNA (mRNA) and translational silencing [1,2]. They have many biological roles, including repression of gene expression by interacting with their target messenger mRNAs and blockade of the translation process or initiate cleavage [2,3]. On that account, they carry the potential to regulate multiple aspects of cellular activity, including differentiation, proliferation, and metabolism. They are also involved in several pathological processes, including viral infection, tumorigenesis, cell death, and serve as biomarkers in many human diseases [1,2,3].

The mammalian microRNAs (miR) are mainly transcribed by Polymerase II (Pol II), resulting in the formation of capped, more than 1000 nt long polyadenylated primary transcripts (pri-miRNAs). The pri-miRNAs contain hairpin-like structures, 60–120 nt long precursor microRNA (pre-miRNA) intermediates, which are processed by RNase III type enzyme Drosha in the nucleus [2]. The pre-miRNAs bearing 2 nt 3’ overhang are subsequently transported to the cytoplasm mediated by Exportin 5, which can bind pre-miRNAs specifically in vitro but only in the presence of Ran-GTP(guanosine triphosphate) cofactor [3]. In the cytoplasm, pre-miRNAs are processed by a second RNase III member protein “Dicer” that cleaves pre-miRNAs to give 22 nt long mature miRNAs. The miRNAs are then incorporated into the RNA-induced silencing complex (RISC) and act as a guide RISC to mediate appropriate target degradation and or translation suppression.

Bioinformatics analyses indicate that a specific miRNA can regulate the expression of several mRNAs through the miRNA-mRNA association, and a particular mRNA can be controlled by multiple miRNAs. MiRNAs may also be regulated by potential targets in vivo [2,4,5,6]. The miRNA-mRNA association suggests that specific regulatory patterns could exist between the coding (mRNAs) and non-coding (miRNAs) RNA molecules. MiRNAs negatively regulate their target mRNAs through complementary binding. Some miRNAs can interact with the 3′ untranslated (3′ UTR) region of their target mRNA and reduce the level of mRNA expression [2,4,5,6]. There are different approaches to identify the miRNA-mRNA interactions, including sequence-based complementarity matches. Computational methods have also included thermodynamic features, such as stability of the miRNA-mRNA heteroduplex and the accessibility of the mRNA target region to identify functional miRNA binding sites [2,4,5,6].

Recently, the role of the bifunctional APOL1-miR193a axis in the transition of glomerular parietal epithelial cells as well as in the maintenance of differentiated phenotype of glomerular visceral epithelial cells (podocytes) has been proposed [7,8]. MicroRNA (miR) 193a can be generated from the 5p or 3p arms of pre-miRNAs. MiR193a-5p plays a vital role in the differentiation of glomerular epithelial cells during embryogenesis [8]. Both glomerular parietal epithelial cells (PECs) and visceral epithelial cells (podocytes) originate from the same mesenchymal cells during embryogenesis [9]. Notably, the expression of miR193a-5p determines whether cells would display PECs’ or podocytes’ phenotype. Putative studies suggested that miR193a binds to mRNA of the APOL1 3’ UTR region [7]. An increase in miR193a expression is associated with an upregulation of the expression of PECs markers, whereas, a decreased expression of miR193a enhances the molecular markers of podocytes [7]. Interestingly, the downregulation of miR193a also induces the expression of APOL1 that participates in the maintenance of the podocyte molecular phenotype [7,8]. On that account, modulation of miR193a expression could be used to preserve APOL1 expression as a therapeutic tool in diseases associated with dedifferentiation of podocytes [10,11,12,13]. However, the modulation of miR193a expression through the interaction with pertinent activators and repressors was not reported.

In in vitro studies, undifferentiated podocytes display enhanced expression of miR193a but lack of podocyte markers, whereas differentiated podocytes exhibit attenuated expression of miR193a but a robust expression of podocyte markers [11]. There is a possibility that transcription factors regulating cellular differentiation may also be contributing to the modulation of miR193a expression. Therefore, we have investigated the relationship between miR193a and differentiating factors such as Yin Yang 1 (YY1), Wilms tumor type 1 (WT1), Sex determining region Y (SRY)-Box2 (Sox2), and Vitamin D Receptor (VDR)/Retinoid X Receptor (RXR) [14,15].

In the present study, the interactions with miR193a were investigated with an integrative analysis of the miR193a promoter region and various transcription factors binding [16,17]. The identification of DR1, DR3 and DR4 elements on the miR193a gene suggested the binding sites for nuclear receptors such as Peroxisome proliferator-activated receptor (PPAR), VDR, and RXR which bind to these elements as dimer or heterodimers [18,19]. The motif scan of the miR193a gene provided information about other transcription factors, including YY1, WT1, and Sox2, which can potentially bind to the miR193a promoter region and may regulate miR193a expression [19,20]. All structure models of transcription factors, including YY1, WT1, Sox2, and VDR-RXR heterodimer [21] and the tertiary structures of the miR193a promoter region were generated [22] and refined using computational tools [23]. The DNA-protein complexes of the miR193a promoter region and different transcription factors were generated using a docking approach [24]. To validate the role of these factors in the transcription of miR193a in human podocytes, profiles of YY1, WT1, Sox2, and VDR were analyzed in undifferentiated podocytes expressing enhanced miR193a and differentiated podocytes expressing attenuated miR193a. To confirm a causal relationship between miR193a and molecular markers of podocytes, we examined the effect of a specific inhibitor of miR193a as well as transfection of miR193a plasmid on the expression of podocyte molecular markers. To validate a cause and effect relationship, the effect of silencing of YY1, WT1, Sox2, and VDR was evaluated on the expression of miR193a, APOL1, and podocyte markers. To analyze the composition of WT1 and VDR-RXR complexes, we analyzed the components of IP (immunoprecipitated) fractions harvested from cellular lysates treated with either WT1 or RXR antibodies. To study the interaction between miR193a and APOL1 mRNA, we have used multiple Bioinformatics approaches, including the pairing of miR193a seed region and complementary sites within APOL1 mRNA [4,5,6,25]. Since the secondary structure components of mRNA and miRNA heterodimer can influence the accessibility of the target site and binding ability of miRNAs, we studied the secondary structural components of APOL1 3’ UTR and miR193a-5p [26,27]. We also studied the tertiary structure of APOL1 3’ UTR and miR193a complex as the thermodynamic properties of the interaction interface is an important parameter to analyze the miRNA-mRNA interactions [28,29,30]. DNA-protein interactions and protein-protein interactions have also been analyzed [31,32,33]. All the structural models and the interactions were visualized [34,35]. To validate the interactions, we have also performed Molecular Dynamic (MD) simulations of the miR193a-APOL1 3’ UTR and DNA-protein complexes [36,37]. To confirm the therapeutic role of miR193a modulation on the preservation of the podocyte molecular phenotype, we examined the effect of VDR agonist (VDA) on the modulation of miR193a and podocyte differentiation molecular marker in undifferentiated podocytes.

## 2. Methods

### 2.1. Human Podocytes

Human podocytes (PDs) were conditionally immortalized by introducing temperature-sensitive SV40-T antigen by transfection [38]. These cells proliferate at the permissive temperature (33 °C) and enter into growth phase arrest after transfer to the non-permissive temperature (37 °C). The growth medium contains RPMI 1640 supplemented with 10% fetal bovine serum (FBS), 1x Pen-Strep, 1 mM L-glutamine, and 1x ITS (Insulin, Transferrin, and Selenium, Invitrogen, Thermo Fisher, Waltham, MA, USA). Undifferentiated podocytes (UPDs; cultured at 33 °C) were seeded on collagen-coated plates and differentiated for 10 days at 37 °C (differentiated podocytes, DPDs). DNA sequencing of these podocytes revealed the endogenous APOL1G0 genotype.

### 2.2. Silencing of YY1, Sox2, and WT1

UPDs/DPDs were transfected with scrambled siRNA (control), siRNA-YY1 (20 nM; Santa Cruz Biotechnology, Dallas, TX, USA), siRNA-Sox2 (20 nM; Santa Cruz Biotechnology, Dallas, TX, USA), or siRNA-WT1 (25 nM; Santa Cruz Biotechnology, Dallas, TX, USA) with Lipofectamine RNAiMAX transfection reagent according to the manufacturer’s protocol (Thermo Fisher, Waltham, MA, USA). Briefly, UPDs/DPDs were transfected at 60–80% confluence in six-well plates. Lipofectamine reagent (9 µL) and siRNAs (10 µM, 2–3 µL) were diluted in opti-MEM media (150 µL) (Thermo Fisher, Waltham, MA, USA). Then, diluted siRNA (150 µL) was added to diluted Lipofectamine reagent (150 µL) in a 1:1 ratio (v/v) and incubated for 5 min at room temperature (25 °C). After incubation, the siRNA lipid-complex was added to cells and kept at 37 °C in opti-MEM media for 48 hrs. The cells were harvested for protein and RNA analyses. Control and transfected cells were used under control and experimental conditions.

### 2.3. Transfection of a Specific miR193a Inhibitor and miR193a Expression Plasmid

A specific miR193a inhibitor (25 nM; cat. no. 4464084; Thermo Fisher Scientific, Waltham, MA, USA), miR193a expression plasmid (25 nM; cat. no. SC400232; OriGene, Rockville, MD, USA), and empty vector (25 nM; pCMV-MIR; OriGene, Rockville, MD, USA) were transfected in the control and experimental cells (UPDs or DPDs) using Lipofectamine 3000 Transfection Reagent (Thermo Fisher Scientific, Waltham, MA, USA) according to the manufacturer’s protocol. Details are described in our recent publications [7,10,11].

### 2.4. RNA Isolation and qPCR Studies

Total RNA was isolated from control and experimental UPDs and DPDs with TRIzol reagent (Invitrogen, Waltham, MA, USA) using an RNA isolation mini kit from Qiagen (C217004) (Germantown, MD, USA).

A 25 µL reaction mix was prepared to contain the SYBR Green reaction mix (Qiagen, Germantown, MD, USA) for all quantitative PCR assays. Real-time PCR was performed following cDNA synthesis [abm’s (Applied Biological Materials, Richmond, BC, Canada) 5x All-In-One RT MasterMix Cat # G486 ] using the cDNA by one-step Universal SYBR Green kit (Qiagen, Germantown, MD, USA) according to the manufacturer’s instructions using specific primers obtained from Thermo Fisher Scientific, USA. The following primers were used:***GAPDH*** fw 5′ CCC ATC ACC ATC TTC CAG GAG 3′; rev 5′ GTT GTC ATG GAT GAC CTT GGC 3′,***WT1*** fw 5′ CGAGAGCGATAACCACACAACG 3′; rev 5′ GTCTCAGATGCCGACCGTACAA 3′,***APOL1*** fw 5′ ATCTCAGCTGAAAGCGGTGAAC 3′; rev 5′ TGACTTTGCCCCCTCATGTAAG 3′,***CD2AP*** fw 5′ CTGTCAGCTGCAGAGAAGAAA 3′; rev 5′ TTGGGTTGGAGAATGTCCAC 3′.***Nephrin*** fw 5′CCCCTCTATGATGAAGTACAAATGGA3′; rev and 5′GTACGGATTTCCTCAGGTCTTCT3′***VDR*** fw 5′-CTTCAGGCGAAGCATGAAGC-3′; rev 5′-CCTTCATCATGCCGATGTCC-3′

Conditions were as follows: 50 °C for 10 min 95 °C for 1 min, followed by 40 cycles of 95 °C for 15 s, 60 °C for 1 min. Quantitative PCR was performed using an ABI Prism 7900HT sequence detection system, and the relative quantification of gene expression was calculated using the ΔΔCT values. Data were expressed as relative mRNA expression in reference to the control, normalized to the quantity of RNA input by performing measurements on an endogenous reference gene (*GAPDH*).

### 2.5. MicroRNA Assay

For miRNA quantification, the total RNA was isolated from control and experimental UPDs/DPDs with RNA isolation kit (as above) and 1 µg of RNA was reverse transcribed using miR193a and U6-snRNA specific RT primers to generate first-strand cDNA from RNA, using abm’s (Applied Biological Materials Inc, Richmond, BC, Canada) miRNA cDNA synthesis kit with Poly (A) polymerase tailing (Cat #G902), according to manufacturer’s instructions. For cDNA, a 25 µL PCR reaction was prepared to contain (1 µg of RNA, and All in one RT Master mix). The PCR condition was as follows: 25 °C for 10 min, 42 °C for 50 min, 85 °C for 5 min, and 4 °C until stopped. Real-time PCR was performed by using the Qiagen SYBR Green PCR master mix and detection primers miR-193a and U6-snRNA (Thermo Fisher, Waltham, MA, USA) in ABI-7500, Applied Biosystems. For real-time PCR a 25 µL reaction mix was prepared to contain SYBR Green PCR master mix II (total volume 25 µL), that includes cDNA (2 µL), nuclease-free water (2 µL), and primer (1 µL). The qPCR conditions were as follows: 50 °C for 2 min 95 °C for 10 min, followed by 40 cycles of 95 °C for 15 s, 60 °C for 1 min. U6 was used as an internal control. Relative quantification of gene expression was calculated using the ΔΔCT values, and the obtained values were normalized to internal control, U6-snRNA expression.

### 2.6. Western Blot Studies

Western blot studies were carried out as described previously [10,11,12]. Briefly, control and experimental cells were harvested, lysed in RIPA buffer containing 50 mM Tris-Cl (pH 7.5), 150 mM NaCl, 1 mM EDTA, 1% NP-40, 0.25% Deoxycholate, 0.1% SDS, 1X protease inhibitor cocktail (Calbiochem, Sigma Aldrich, St. Louis, MO, USA Cocktail Set I), 1 mM Phenylmethylsulfonyl fluoride (PMSF), and 0.2 mM sodium orthovanadate. Protein concentration was determined using the Biorad (Hercules, CA, USA) Protein Assay kit. Total protein lysed extracts (15–30 mg/lane) were loaded on a 10% polyacrylamide (PAGE) and, after gel electrophoresis, transferred onto Polyvinylidene fluoride (PVDF) membranes were processed for immunostaining with primary antibodies against YY1 (mouse monoclonal, 1:1000, #SC 7341; Santa Cruz Biotechnology, Dallas, TX, USA), Sox2 (mouse monoclonal, 1:3000, #SC 365823; Santa Cruz Biotechnology, Dallas, TX, USA), APOL1 (mouse monoclonal, 1:1000, Protein tech, Rosemont, IL, USA), WT1 (rabbit polyclonal, 1:1000, Abcam, Cambridge, MA, USA), nephrin (rabbit polyclonal #ab58968; Abcam, Cambridge, MA, USA), CD2AP (rabbit polyclonal; #SC 25272; Santa Cruz Biotechnology, Dallas, TX, USA), GAPDH (mouse monoclonal, 1:15000, #SC 166515; Santa Cruz Biotechnology, Dallas, TX, USA), VDR (mouse monoclonal, 1:1000, #SC 13133; Santa Cruz Biotechnology, Dallas, TX, USA), SMRT (silencing mediator for thyroid and retinoid hormones; mouse monoclonal, 1:1000, #SC 32298; Santa Cruz Biotechnology, Dallas, TX, USA), HDAC1 (mouse monoclonal, 1:1000, # 05-100-I, Millipore Sigma, Sigma Aldrich, St. Louis, MO, USA), HDAC3 (Histone deacetylase 3; rabbit polyclonal, 1:3000, #SC38155, Santa Cruz Biotechnology, Dallas, TX, USA), RXR (rabbit polyclonal, 1:1000, #SC831, Santa Cruz Biotechnology, Dallas, TX, USA), EZH2 (rabbit polyclonal, 1:1000, # 07-1562, Millipore Sigma, Sigma Aldrich, St. Louis, MO, USA), and H3K27me3 (rabbit monoclonal, 1:3000, #9733P, Cell Signaling Technology, Danvers, MA, USA) followed by treatment with horseradish peroxidase-labeled appropriate secondary antibodies (1;3000, Cell Signaling Technology, Danvers, MA USA). Equal protein loading and the protein transfers were confirmed by immunoblotting for determination of GAPDH protein using mouse monoclonal GAPDH antibody (1:15,000 #SC47724; Santa Cruz Biotechnology, Dallas, TX, USA) on the same (stripped) Western blots. The blots were developed using a chemiluminescence detection kit (PIERCE, Rockford, IL, USA) and scanned on the Bio-Rad (Hercules, CA, USA) ChemiDoc MP imaging system (Biorad, Hercules, CA, USA) using Image Laboratory software. The images captured were then converted to either a TIFF or JPEG format and were further subjected to Adobe Photoshop/Microsoft PowerPoint software. For quantification, we used ImageJ software (NIH, Bethesda, MD, USA), and further analyses were performed on a Microsoft Excel Sheet/ Graph pad software (California corporation, La Jolla, CA, USA).

### 2.7. Immunoprecipitation (IP)

Lysates from undifferentiated (UPDs) and differentiated PDs (DPDs) were first immunoprecipitated following the addition of either 5 μg of RXR (Santa Cruz, Biotechnology, Dallas, TX, USA) or 5 μg of WT1 (Abcam, Cambridge, MA, USA). The immune complexes were then collected using 25 μL of Protein-A + G sepharose beads (GE Health Care, Life Sciences, Marlborough, MA, USA) in Radioimmunoprecipitation assay (RIPA) buffer. IP was carried out at 4 °C, for 4 h, on a rotating platform. Following this, precipitated A/G proteins were pelleted down by centrifugation at 4500 rpm for 10 min at 4 °C. Next, the protein pellet was washed (3X) each time with 1 mL of cold RIPA lysis buffer followed by centrifugation each time for 10 min at 2500 rpm in a microfuge. After washings, beads were re-suspended in 100 μL of lysis buffer to which (sodium dodecylsulfate polyacrylamide gel electrophoresis (SDS-PAGE) sample buffer (50 μL) was added, and samples were heated at 100 °C, followed by SDS-PAGE and immunoblotted using specific antibodies as indicated.

### 2.8. miR193a Gene and Transcription Factor Binding

We identified the human miR193a gene using NCBI BLAST [16] and promoter as well as transcription factor binding was searched using NUBIScan [17], TFsitescan [18], PROMO [19] and miRMap [25] tools. The NUBIScan uses nucleotide distribution matrices to identify nuclear receptor (NR) binding sites. The Multiple EM for Motif Elicitation (MEME) Motif identification tool [20] was used to identify specific motifs in the miR193a gene.

### 2.9. Homology Modeling and Docking Studies

We have used the docking approach to provide the binding conformation of transcription factors to the miR193a promoter. It also provides information about nucleotide interacting residues and the features of the DNA-protein interaction interface.

We used the ITasser tool [21] for a template-based modeling approach to generate the structure of different proteins and docking approaches to generate miR193a and transcription factor binding complexes [24]. The 3-dimensional structure model of miR193a promoter or gene was created using the 3D DART tool [22], and structure models of different transcription factors, including YY1, WT1, SOX2, and VDR-RXR heterodimer were refined using Galaxy Refine [23]. The refinement process is based on repetitive relaxations by short molecular dynamics simulations for mild (0.6 ps) and aggressive (0.8 ps) relaxations with 4 fs time step after structure perturbations. The refinement of models improved specific parameters, for example, an increase in Rama favored residues and decrease in poor rotamers [23]. The DNA-protein complexes were generated using ZDock tool [24], which uses the Fast Fourier Transform algorithm to enable an efficient global docking search on a 3D grid, and utilizes a combination of shape, complementarity, electrostatics and potential statistical terms for scoring.

### 2.10. DNA-Protein Interactions

We analyzed the interaction between transfection factors and miR193a promoter to obtain information about amino acid residues, which could be involved in signal transduction and to be validated in experimental conditions.

The thermodynamic properties were analyzed using the PDBePISA tool [31]. We used DNAPRODB [32] and KFC2 [33] tools to identify the interaction of miR193a and different transcription factors. 

### 2.11. Visualization

All the 3-D structures of protein-protein and DNA-protein complexes were visualized, and figures were generated using PyMol [34] and Chimera [35] tools.

### 2.12. APOL1 3′UTR and miR193a Sequence Identification

We identified the APOL1 3′ UTR and miR193a-5p sequences using TargetScan [5,6], mirbase [4], and miRMap [25] tools.

### 2.13. APOL1 3′ UTR and miR193a Secondary Structure

The secondary structures of APOL1 3’ UTR and miR193a provide information about which conformation has the lowest energy in the hybrid structure. The hybrid structure can have multiple conformations based on the possible interaction between mRNA and miRNA bases, and these secondary structure elements play a significant role in the seed matching and pairing of bases. The overall conformation of the hybrid structure determines the efficacy of miRNA induced regulation of target mRNA.

We generated the APOL1 3′ UTR and miR193a secondary structures using RNAStructure [26] and mfold [27] tools.

### 2.14. APOL1 3′ UTR and miR193a Tertiary Structure Prediction and Docking

We generated the APOL1 3’ UTR and miR193a tertiary structures using SimRNA [28] and RNAComposer [29] tools. APOL1 3’ UTR and miR193a docking were performed using HNADOCK [30] tool.

### 2.15. Molecular Dynamics (MD) Simulations

Molecular Dynamics (MD) Simulations of the complexes provide information about the stability, and the Root means square deviation (RMSD) of complexes during the course of simulation and the Root mean square fluctuation (RMSF) of individual residues after 5 ns and 10 ns of simulation. The Radius of gyration (Rg) values shows the variation of structures during the course of the simulation.

MD simulations of APOL1-3′ UTR-miR193a and VDR-RXR, Sox2, YY1, and WT1 complexed with miR193a promoter were performed using the GROMACS 2019.2 package [36,37]. The structures were solvated in an explicitly represented TIP4P water box, extending 1 nm from the protein in each direction. NaCl was added to each system to neutralize the charge of each protein and produce a final physiological salt concentration of 150 mM. Solvation and ionization of the system were performed using GROMACS (GROningen MAchine for Chemical Simulations).

The systems were relaxed by energy minimization using the steepest descent method to remove strain in the initial structure. All bonds involving hydrogen were constrained using the matrix based LINCS algorithm, and the systems were subjected to an NVT (constant number of particles, volume, and temperature) equilibration for 500 ps to 325 K at 1 bar, followed by an NPT (regular number of particles, pressure, and temperature) equilibration for 500 ps to 1 bar at 325 K.

MD simulations were done using the Amber99SB-ILDN force field in the GROMACS 2019.2 package with a time step of 2 fs, and trajectories were recorded every 10 ps. A cutoff distance of 1.2 nm was used for non-bonded calculations, with a smooth switching function at 1.0 nm. The standard particle-mesh Ewald method was used to calculate the long-range electrostatic interactions. All bonds involving hydrogen were constrained using the LINCS algorithm during equilibration and production runs. All simulations were conducted under isothermal and isobaric conditions using an NPT ensemble by maintaining the temperature at 325 K using the V-rescale modified Berendsen thermostat method and keeping the pressure at 1 atm using the Parrinello-Rahman pressure coupling method.

Analysis of MD simulations was done using the gmx rms, gmx fmsf, and gmx gyrate distribution programs, and data were plotted using XMGrace. Backbone RMSD, per-residue RMSF, and radius of gyration values were calculated by extracting trajectories from each simulation at different times (from 5 ns and 10 ns).

### 2.16. Statistical Analyses

Statistical comparisons were performed with the Prism program using the Mann–Whitney *U* test for nonparametric data and the unpaired *t*-test for parametric data. A *p* value *<* 0.05 was considered statistically significant.

## 3. Results

### 3.1. Evaluation of Molecular Profiles of Undifferentiated (UPD) and Differentiated Podocytes (DPD)

To determine the expression profile of miR193a of undifferentiated (UPDs) and differentiated podocytes (DPDs), RNAs were extracted from four independent cellular lysates of UPDs (the cultured podocytes at 33 °C) and DPDs (the cultured podocytes at 37 °C for 10 days) and assayed for miR193a. As shown in Figure 1A, DPDs showed five-fold downing (*p* < 0.05) of miR193a when compared to UPDs.

To evaluate protein expression profiles of UPDs and DPDs, proteins were extracted from four independent cellular lysates for UPDs and DPDs. Protein blots were probed for podocyte molecular markers (CD2AP and WT1) and cellular differentiating transcription factors (VDR, YY1, and Sox2). The protein blots were reprobed for GAPDH. Gels are displayed in Figure 1B. The same cellular lysates were also probed for nephrin and APOL1 (human podocyte markers) and reprobed for GAPDH. Gels are shown in Figure 1C. Densitometric data are shown in the form of dot plots in Figure 1D–J). DPDs displayed enhanced (*p* < 0.05) expression of podocyte markers (CD2AP, WT1, and Nephrin) and VDR but attenuated expression of YY1 and Sox2. On the other hand, YY1 and Sox2 expressions were increased in UPDs, the cells exhibiting escalated expression of miR193a. Since differentiated podocytes (DPDs) displayed enhanced expression of WT1 and VDR but attenuated expression of miR193a, we concluded that these molecules could be forming repressor complexes on miR193a promoters. In contrast, undifferentiated podocytes (UPDs) not only showed enhanced expression of miR193a but also of YY1 and Sox2. Therefore, we presumed that both YY1 and Sox2 could be acting as activators on the miR193a promoter.

### 3.2. Prediction of Binding Sites for Transcription Factors

The miR193a gene was identified using the NCBI BLAST tool [16] that provided hits related to human miR193a (NR_029710.1) with expecting value (E value 5 × 10^−38^) and percent identity 100% and Homo sapiens chromosome 17, GRCh38.p12 Primary Assembly (NC_000017.11) with the expected value (E value 1 × 10^−138^) and percent identity 99.63%. The miR193a gene (NC_000017.11) was analyzed for transcription factors, and the results suggested that various transcription factors, including nuclear receptors, can bind to the miR193a gene. The NUBIScan tool [17] provided information about direct repeats (DRs), palindromic inverted repeats (IRs), everted repeats (ERs) of single DNA hexamer which are present in a distinct arrangement. The majority of nuclear receptors (NRs) binds as homo- or heterodimers to these hexamer repeats, each partner of the dimer interacts with one of the hexamers. The peroxisome proliferator-activated receptor (PPAR) and retinoid x receptor (RXR) dimer would prefer binding at DR1 element at 39 (+) strand (GGTGCAcAGAGCC). The liver x receptor (LXR) and retinoid x receptor (RXR) dimer may prefer binding at DR4 element at 174 (–) strand (GGGACTttgtAGGCCA), 29 (+) strand (AGCGGAgcgcGGTGCA), and 214 (–) strand (AGGACCggggAGAAGA). The vitamin D receptor (VDR) and retinoid x receptor (RXR) dimer can also bind at DR3 element at 7 (+) strand (GGGACAcccAGAGCT). We identified YY1 (5–8, ATGG), RXR alpha (21–27, GGGTCTT and 42–48, GGGTGTC), PPAR alpha, and RXR alpha (17–27, GGCTGGGTCTT and 66–76, AAGTCCCAGTT), SOX2, WT1, and nuclear receptors VDR-RXR binding sites on the miR193a gene (Figure 2A). The MEME Motif [20], JASPARv2020 [39] and STAMP [40] tools identified Sox2 motif in the miR193a gene (Figure 2B).

### 3.3. Evaluation of the Transcription Factors Enhancing miR193a Expression and Associated Downstream Signaling

The analyses of transcription factors and miR193a promoter complexes suggested binding of YY1 and Sox2 at miR93a promoter as activating factors. Information about the thermodynamic properties of the interaction and nucleotide interacting residues are displayed in Table 1 and Table 2. The thermodynamic properties of transcription factors and miR193a were analyzed in terms of ΔG^int^ that is solvation free energy gain upon the formation of the assembly in Kcal/mol, which is calculated as the difference in total solvation energies of isolated and assembled structures. This value does not include the effect of satisfying hydrogen bonds and salt bridges across the assembly interfaces. The ΔG^diss^ indicates the free energy of assembly dissociation in Kcal/mol, and TΔS^diss^ shows the rigid body entropy change at dissociation in Kcal/mol.

The thermodynamic properties of miR193a and YY1 complex suggested that the overall surface area is 2745.9 Å^2^. ΔG^int^ that indicates the solvation free energy gain upon the formation of the assembly in kcal/mol is −50.4 kcal/mol. The value of ΔG^diss^, which shows the free energy of assembly dissociation in kcal/mol, is 107.3 kcal/mol. The free energy of dissociation corresponds to the free energy difference between dissociated and associated states. Positive values of ΔG^diss^ indicate that an external driving force should be applied to dissociate the assembly; therefore, the assemblies with ΔG^diss^ > 0 are thermodynamically stable. The rigid body entropy change at dissociation TΔS^diss^ in kcal/mol is 14.2 kcal/mol. The number of hydrogen bonds and salt bridges in miR193a and YY1 complex suggested that there are 160 hydrogen bonds and no salt bridge formation. The symmetry number that indicates the number of different but equivalent orientations of the assembly, which can be obtained by rotation, its value is 1.

The thermodynamic properties of miR193a and Sox2 complex suggested that the overall surface area is 3165.8 Å^2^. ΔG^int^ that indicates the solvation free energy gain upon the formation of the assembly in kcal/mol is −58.1 kcal/mol. The value of ΔG^diss^, which indicates the free energy of assembly dissociation in kcal/mol, is 115.4 kcal/mol. The free energy of dissociation corresponds to the free energy difference between dissociated and associated states. Positive values of ΔG^diss^ indicate that an external driving force should be applied to dissociate the assembly; therefore, the assemblies with ΔG^diss^ > 0 are thermodynamically stable. The rigid body entropy change at dissociation TΔS^diss^ in kcal/mol is 14.2 kcal/mol. The number of hydrogen bonds and salt bridges in miR193a and Sox2 complex suggested that there are 161 hydrogen bonds and no salt bridge formation. The symmetry number that indicates the number of different but equivalent orientations of the assembly, which can be obtained by rotation, its value is 1.

To establish a causal relationship between enhanced YY1/Sox2 expressions and enhancement of miR193a levels, we determined the expression of miR193a in siRNA-YY1- and Sox2-silenced UPDs. UPDs were transfected with siRNA-YY1 or siRNA-Sox2. miR193a were assayed in control, siRNA-YY1- and siRNA-Sox2-transfected cells. The results of four independent sets are shown in Figure 3C. Silencing of YY1 or Sox2 downed miR193a expression by 3-fold.

We asked if silencing of YY1 and Sox2 decreasing the expression of miR193a in UPDs, then it should be associated with de-repression of mRNA expression of *WT1*, *CD2AP*, *Nephrin*, and *VDR*. To examine the downstream-signaling effect of the downing of miR193a, cDNAs were prepared from the RNAs from the above-mentioned cellular lysates and amplified with specific primers for *YY1*, *Sox2*, *VDR*, *WT1*, *CD2AP*, and *Nephrin*. Cumulative data are shown in bar graphs (Figure 3D). UPDs silenced for either YY1 or Sox2 showed enhanced expression of *WT1*, *CD2AP*, *Nephrin*, and *VDR*.

If YY1 and Sox2 were modulating the expression of miR193a in podocytes, their silencing should alter the molecular phenotype of undifferentiated podocytes (UPDs). To validate this notion, UPDs were transfected with either scrambled (SCR), siRNA-YY1, or siRNA-Sox2. Protein blots were probed for YY1, Sox2, CD2AP, WT1, VDR, and GAPDH from three independent cellular lysates. Representative gels from two different lysates are shown in Figure 3E. Cumulative densitometric data are shown in Figure 3F. UPDs silenced for either YY1 or Sox2 showed an increase in the expression of podocyte differentiating markers.

### 3.4. Determination of Repressors Contributing to the Downing of miR193a Expression

The analyses of transcription factors and miR193a promoter complexes suggested binding of VDR-RXR and WT1-EZH2 at miR93a promoter as repressors. Details about thermodynamic properties of the interaction and nucleotide interacting residues are displayed in Table 1; Table 2.

The miR193a and VDR-RXR heterodimer complex suggested that the DNA binding domain (DBD) of RXR mainly interacts with the miR193a promoter while the substrate-binding domain (SBD) may also be responsible for this interaction (Figure 4A).

We asked whether VDR-RXR heterodimer is acting as a repressor for downing of miR193a in DPDs. DPDs were transfected with either siRNA-VDR or siRNA-RXR. RNAs were extracted and assayed for miR193a. Cumulative data from three independent experiments are shown in Figure 4B. The silencing of either VDR or RXR increased the expression of miR193a by 4-fold. These findings suggested a causal relationship between VDR/RXR expression and downing of miR193a.

The thermodynamic properties of miR193a and WT1 complex suggested that the overall surface area is 2812.8 Å^2^. ΔG^int^ that indicates the solvation free energy gain upon the formation of the assembly in kcal/mol is −52.2 kcal/mol. The value of ΔG^diss^, which shows the free energy of assembly dissociation in kcal/mol, is 109.5 kcal/mol. The free energy of dissociation corresponds to the free energy difference between dissociated and associated states. Positive values of ΔG^diss^ indicate that an external driving force should be applied to dissociate the assembly; therefore, the assemblies with ΔG^diss^ > 0 are thermodynamically stable. The rigid body entropy change at dissociation TΔS^diss^ in kcal/mol is 14.2 kcal/mol. The number of hydrogen bonds and salt bridges in miR193a and WT1 complex suggested that there are 161 hydrogen bonds and no salt bridge formation. The symmetry number that indicates the number of different but equivalent orientations of the assembly, which can be obtained by rotation, its value is 1.

To establish a causal relationship between enhanced expression of WT1 and downing of miR193a, DPDs were transfected with siRNA-WT1. RNAs were extracted from three cellular lysates of control and siRNA-WT1 transfected DPDs and assayed for miR193a. Cumulative data from three independent experiments are shown in Figure 4D. The silencing of WT1 escalated the expression of miR193a by 5-fold.

### 3.5. Molecular Dynamics (MD) Simulation of miR193a Promoter and Transcription Factors

#### 3.5.1. YY1 and miR193a

The MD simulation of the miR193a-YY1 complex suggested that the complex structure is more stable after 10 ns and converges with a root mean square deviation (RMSD) around 0.7 nm. The radius of gyration (Rg) shows a minimal deviation in the DNA bound form of YY1. The root mean fluctuation (RMSF) values of some of the interacting residues show fluctuations during the course of the simulation, but the overall stability increases upon binding with the DNA (Appendix A).

#### 3.5.2. Sox2 and miR193a

The MD simulation of the miR193a-Sox2 complex suggested that the complex is a stable structure with a root mean square deviation (RMSD) around 1.2 nm. The DNA bound Sox2 shows a little fluctuation initially during 2–4 ns of simulation. The root mean square fluctuation (RMSF) values of some of the interacting residues show fluctuation upon binding to the miR193a promoter (Appendix A).

#### 3.5.3. VDR-RXR and miR193a

The MD simulation of miR193a and VDR-RXR heterodimer suggested that the complex is very stable after 10 ns simulation and converges around root mean square deviation (RMSD) value 0.7 nm. The RXR-DNA binding domain residues are mainly interacting with the DNA. The miR193a and VDR-RXR heterodimer complex show a little fluctuation after 6 ns of simulation in the range of 2.9–3.0 nm of the radius of gyration (Rg) values. The root mean square fluctuation (RMSF) values of the interacting residues in the DNA binding domain (DBD) of RXR and VDR-DBD after 5 ns and 10 ns are shown in Appendix A.

#### 3.5.4. WT1 and miR193a

The MD simulation of the miR193a-WT1 complex suggested that the complex is a very stable structure, and the root mean square deviation (RMSD) value converges around 1.2 nm. The DNA bound WT1 shows the constant radius of gyration (Rg) during the course of the simulation. The root mean square fluctuation (RMSF) values of some of the interacting residues show some fluctuation upon interaction (Appendix A).

### 3.6. Analysis of VDR-RXR Repressor Contributing to the Downing of miR193a Expression

The interaction between VDR-RXR, SMRT, and HDAC3 suggested that the free energy gain of complex formation (ΔG^i^) is −48.6 kcal/mol and the free energy of dissociation ΔG^diss^ is −7.3 kcal/mol. The rigid body entropy change at dissociation TΔS^diss^ in kcal/mol is 23.7 kcal/mol. A negative value of ΔG^diss^ indicates that the association of the repressor complex is transient. The repressor complex shows that SMRT interacts strongly with RXR (Interface area 847.3 Å^2^) with a free energy of association (ΔG^int^) −12.4 kcal/mol and forms 4 hydrogen bonds. The hydrogen bonds include Gln275 and Ala455 (3.88 Å), Lys279 and Lys449 (3.79 Å), Met459 and Arg426 (2.35 Å) and Thr333 and Leu453 (3.57 Å). The interaction between RXR and HDAC3 (Interface area 873.8 Å^2^) suggested that RXR interacts with HDAC3 with a free energy gain on assembly formation (ΔG^i^) −9.8 kcal/mol and forms 3 hydrogen bonds. The hydrogen bonds include Thr271 and Tyr18 (3.58 Å), Asn340, and Thr42 (3.35 Å), Phe443, and His22 (3.83 Å). The interaction between SMRT and HDAC3 (Interface area 541.1 Å^2^) suggested that SMRT interacts with HDAC3 with a free energy gain on assembly formation (ΔG^i^) −6.6 kcal/mol and forms 2 hydrogen bonds and 1 salt bridge. The hydrogen bonds include Glu459 and His17 (3.09 Å) and Arg460 and Lys25 (3.31 Å). The Glu459 forms a salt bridge (3.09 Å) with His17. The structural construct of VDR-RXR is shown in Figure 5A.

We proposed that VDA bound VDR makes a heterodimer with RXR and recruits SMRT (a nuclear co-repressor) and forms a repressor complex with HDAC3, as shown in the schematic diagram (Figure 5B).

To explore the components of VDR-RXR repressor complex, protein blots of three different cellular lysates of DPDs were probed for RXR, VDR, SMRT, HDAC3, and GAPDH. Gels of three independent lysates are displayed in Figure 5C. Cumulative densitometric data are shown as dot plots in Figure 5D–G. DPDs exhibited higher expression of RXR and VDR when compared to UPDs.

To validate the components of VDR-RXR complexes, above mentioned lysates, were immunoprecipitated with the RXR antibody. Protein blots of RXR-IP fractions were probed for RXR, VDR, SMRT, HDAC3, and IgG. Gels are displayed in Figure 5H. Cumulative densitometric data are shown in Figure 5I–L. IP fractions showed enhanced expression of RXR, VDR, SMRT, and HDAC3. These findings suggested that the VDR-RXR repressor complex includes SMRT and HDAC3.

### 3.7. Analysis of WT1 Repressor Complex Contributing to the Downing of miR193a

The binding interactions between WT1-EZH2 and HDAC1 suggested that the free energy gain of complex formation (ΔG^i^) is −30.0 kcal/mol, and the free energy of dissociation ΔG^diss^ is 4.7 kcal/mol. The rigid body entropy change at dissociation TΔS^diss^ in kcal/mol is 15.2 kcal/mol. The repressor complex shows that WT1 interacts with EZH2 (Interface area 1590.3 Å^2^) with a free energy of association (ΔG^int^) −14.7 kcal/mol and forms 12 hydrogen bonds. The hydrogen bonds include Tyr48 and Asn360 (3.83 Å), Gly49 and Asn361 (3.88 Å), Ser50 and Ser362 (3.21 Å), Tyr227 and Asn371 (3.71 Å), Thr277 and Ser375 (3.70 Å), Asn9 and Ser412 (3.64 Å), Cys24 and Met420 (2.60 Å), Ala32 and Asn423 (3.77 Å), Gln33 and Arg355 (3.81 Å), Gly44 and Asn361 (2.81 Å), Tyr48 and Arg456 (3.75 Å), Gly53 and Cys414 (2.69 Å). The interaction between EZH2 and HDAC1 (Interface area 1654.0 Å^2^) suggested that EZH2 interacts with HDAC1 with a free energy gain on assembly formation (ΔG^i^) −15.4 kcal/mol and forms 8 hydrogen bonds and 6 salt bridges. The hydrogen bonds include Tyr133 and Asn354 (3.90 Å), Asp140 and Thr351 (3.31 Å), Gln141 and Glu468 (3.60 Å), Gly143 and Thr351 (3.41 Å), Arg161 and Glu235 (3.52 Å), Val138 and Thr351 (3.80 Å), Asp140 and Thr351 (2.88 Å), Glu162 and Ser246 (3.88 Å). The salt bridges include Arg161 and Glu238 that form three salt bridges of lengths 3.70 Å, 3.59 Å, and 3.82 Å, respectively. The residue Arg161 forms another salt bridge with Glu235 (3.52 Å). The residue Asp160 forms two salt bridges with Lys363 of lengths 3.03 Å and 2.32 Å. The structural construct of WT-EZH2 repressor complex is shown in Figure 6A.

We proposed that WT1 binds with the miR193a promoter and recruits the methyltransferase EZH2 along with HDAC1 resulting methylation of K27 at Histone (H) 3 tail, as shown in the schematic diagram 6B.

To analyze the components of WT1 repressor complex, protein blots of three different cellular lysates of DPDs were probed for EZH2, WT1, HDAC1, H3K27me3, and GAPDH. Gels of three independent lysates are displayed in Figure 6C. Cumulative densitometric data are shown as dot plots in Figure 6D–G. DPDs exhibited higher expression of WT1, HDAC1, and H3K27me3.

To validate the other components of WT1 repressor complexes, above mentioned lysates, were immunoprecipitated with the WT1 antibody. Protein blots of WT1-IP fractions were probed for EZH2, WT1, HDAC1, H3K27me3, and IgG. Gels are displayed in Figure 6H. Cumulative densitometric data are shown in Figure 6I–L. IP fractions showed enhanced expression of EZH2, WT1, HDAC1, and H3K27me3.

### 3.8. Comparison of miR193a-5p Targets

We used the miRDB database to select and compare putative miR193a-5p targets. We searched for proteins such as BH3-like motif containing, cell death inducer protein (BLID) with 3’ UTR length 242, Synaptopodin 2 like protein (SYNPO2L) with 3’ UTR length 1837, NFKB inhibitor interacting Ras-like 1 protein (NKIRAS1) with 3’ UTR length 1032, G-protein coupled receptor 17 (GPR17) with 3’ UTR length 894, Eukaryotic translation initiation factor 2 subunit alpha (EIF2S1) with 3’ UTR length 3076. These proteins have similar motifs or somehow interact with APOL1. The most prevalent APOL1 3’ UTR has 1498 bases, and the other 3’ UTR has 269 bases. We compared the miR193a target repression strength using the miRmap database [25]. The comparison of energy values of miRNA-mRNA binding in the raw value format shows that APOL1 could be an effective target for miR193a-5p. The lower binding energy of miR193a and higher ΔG open, which is the energy required for dissociation of the complex, suggested that APOL1 could be an effective target for miR193a-5p. The seed location should not be at the center of the long UTR in the path of ribosome and translation machinery as the translation machinery could displace the silencing complex. Moreover, the 3’ UTR position, which is the distance from the 3’ UTR end, seems to be ideal for miR193a activity as the distance from the stop codon or the location of seed plays a significant role in miRNA activity (Table 3).

### 3.9. Molecular Genetics of APOL1

APOL1 has 2 locus allele variants APOL1G1 consisting of 2 nonsynonymous coding variants rs73885319 (Ser342Gly) and rs60910145 (Ile384Met), both in the last exon of APOL1. Close to G1, there is another variant APOL1G2 with 6 bp deletion rs71785313 that removed amino acid N388 and Y389. As the proximity of rs73885319, rs60910145, and rs71785313, G1 and G2 are exclusive, recombination between them is very unlikely. The APOL1 SNPs rs73885319 and rs60910145 were identified as risk factors because the 2 missense mutations are in nearly perfect linkage disequilibrium. On a population genetics basis, they can be considered as missense risk haplotypes. The rs73885319 SNP, rs60910145 SNP, and rs71785313 INDEL show 6 different overlapping missense variants that are located on a specific transcript strand with a specific position in the transcript, coding sequence (CDS) and in the protein sequence. For example, one missense variant Ser342Gly on the transcript strand (ENST00000397278.7) is located at 1253 (out of 2918) in the transcript, at 1024 (out of 1197) in the CDS and 342 (out of 398) in the protein sequence, another variant Ile384Met on the transcript strand (ENST00000397278.7) is located at 1381 (out of 2918) in the transcript, at 1152 (out of 1197) in the CDS and 384 (out of 398) in the protein sequence. The inframe deletion in the rs71785313 INDEL of N388 and Y389 on the transcript strand (ENST00000397278.7) is located at 1389–1398 (out of 2918) in the transcript, 1160–1169 (out of 1197) in the CDS and 387–390 (out of 398) in the protein sequence. This transcript strand has several short sequence variants, including 5’ UTR variants, splice region variants, intron variants, start lost variants, missense variants, frameshift variants, synonymous variants, coding sequence variants, splice donor variants, stop gained variants and 3’ UTR variants. Moreover, these variants belong to specific consensus CDS sets coding for APOL1 isoforms, and other possible mutations may be responsible for different disease-related phenotypes.

### 3.10. APOL1 3′ UTR and miR193a Secondary Structures

The most energetically favored conformations of APOL1 3’ UTR and miR193a show secondary structure elements such as External loop, stacks, Helices, bulge loop, Interior loop, and hairpin loops which are characteristics of RNAs. The bimolecular hybrid secondary structures generated by allowing intramolecular interactions and excluding intramolecular interactions predicted conformations with −42.7 and −17.0 kcal/mol energy, respectively (Figure 7A) [27]. The first hybrid structure shows 3’overhang of miR193a, and the second hybrid structure shows intensive pairing of miR193a with APOL1 3’ UTR region.

### 3.11. miR193a Binding with 3′ UTR Region of APOL1

The binding site of miR193a-5p at 3′ UTR region of APOL1 was identified using TargetScan, miRmap [25], and IntaRNA [41]. The TargetScan tool [5,6] provided information about miR193a-5p binding with the most prevalent transcript of APOL1 (ENSG00000100342.16) supported by 3P seq tag (ENST00000422706.1) which is 1498 nt long (Figure 7C). The miR193a-5p binds at position 1027–1033 of site type 7mer-A1 with weighted context ++ score −0.08 and context ++ score percentile 65. Moreover, miR193a-5p also binds to the less prevalent transcript of APOL1 supported by 3P seq tag (ENST00000397279.4), which is 269 nt long. The miR193a-5p binds at position 246–252 of site type 7mer-A1 with weighted context ++ score -0.16 and context ++ score percentile 85. The 7mer-A1 site type comprises the seed (miRNA nucleotides 2–7) match supplemented by an A across from nucleotide 1 of miRNA, supporting the validity of cellular messages that either decrease following miRNA addition or increase following miRNA disruption preferentially contain seed matches with the hierarchy of site efficacy 8mer> 7mer- m8 > 7mer-A1 > 6 mer [5,6]. The miRmap [25] shows that ΔG of binding (binding energy based on ensemble free energy) is −17.90 kcal/mol. The ΔG duplex seed (minimum free energy of the seed with RNAcofold) and ΔG binding seed (binding energy of the seed based on ensemble free energy) is −8.70 kcal/mol and -9.11 kcal/mol respectively. The mRNA opening free energy or accessibility, ΔG open, is 20.37 kcal/mol. The minimum free energy (MFE) with RNAcofold, ΔG duplex, is −16.80 kcal/mol. Moreover, the sum of ΔG open and ΔG duplex or ΔG total is 3.57 kcal/mol. The AU nucleotide composition around the seed or sequence AU content is 0.42. The UTR position, distance from the nearest 3’ UTR end is 456.00, and 3’ compensatory pairing is 2.00. The binding site is at a distance from the stop codons (UAG, UGA, and UAA), and the 3’ pairing score is also significant for the optimal activity of the RISC complex. The IntaRNA [41] suggested the binding energy −7.9 kcal/mol with hybridization energy −16.6 kcal/mol.

### 3.12. Tertiary Structure of APOL1 3′ UTR and miR193a

The tertiary structure of APOL1 3′ UTR and miR193a complex was generated using the sequence and secondary structural information (Figure 7B) [28]. The thermodynamic properties of APOL1 3′ UTR and miR193a-5p (Table 4) suggested that the overall surface area of the interface is 1341.3 Å^2^. ΔG^int^ that indicates the solvation free energy gain upon the formation of the assembly is −27.3 kcal/mol. The value of ΔG^diss^, which shows the free energy of assembly dissociation in kcal/mol, is 29.7 kcal/mol. Positive values of ΔG^diss^ indicate that an external driving force should be applied to dissociate the assembly; therefore, the assemblies with ΔG^diss^ > 0 are thermodynamically stable The rigid body entropy change at dissociation TΔS^diss^ in kcal/mol is 11.8 kcal/mol. In the complex of miR193a and APOL1 3′ UTR complex are predicted 32 hydrogen bonds and no salt bridge. The symmetry number indicates a different number, but equivalent orientations of the assembly can be obtained by rotation; its value is 1 [31].

### 3.13. Evaluation of the Effect of the Downing of miR193a and Overexpression of miR193a on the APOL1 Expression and Podocyte Molecular Phenotype

We asked if miR193a is interacting with APOL1 then, modulation of miR193a expression would inversely regulate the expression of APOL1 as well as markers of podocyte molecular markers. Since UPDs carry a higher expression of miR193a, UPDs were transfected with either control plasmid (empty vector, EV) or specific inhibitor of miR193a (25 nM; plasmid-based) (n = 3). On the other hand, DPDs display attenuated expression of miR193a; therefore, they were transfected with either control plasmid (EV) or miR193a plasmid (50 nm) (n = 3). RNAs were assayed for miR193a. Cumulative data is shown in a bar diagram (Figure 7C). miR193a inhibitor decreased the expression of miR193a by five-fold in UPDs; in contrast, miR93a-transfected DPDs showed a two-fold increase in their miR193 expression (Figure 7C).

cDNAs from EV- and miR193a inhibitor-treated UPDs were amplified with specific primers for *APOL1*, *VDR*, *WT1*, and *Nephrin* (n = 3). Cumulative data are shown in bar graphs (Figure 7D). Downing of miR193a in UPDs increased mRNA expression of *APOL1* and *WT1* by three-fold and of VDR and Nephrin by two-fold.

cDNAs from EV and miR193a-transfected DPDs were also amplified with specific primers for *APOL1*, *VDR*, *WT1*, and *Nephrin* (n = 3). Cumulative data are shown in a bar diagram (Figure 7E). miR193 over-expressing DPDs not only displayed an attenuated expression of APOL1 and VDR but also showed a significant decrease in the expression of *WT1* and *Nephrin* (podocyte molecular markers).

These findings indicate that miR193a negatively regulates the expression of *APOL1*, *VDR*, *WT1*, and *Nephrin*.

### 3.14. Molecular Dynamics (MD) Simulation of miR193a-5p and APOL1 3′ UTR

The Molecular Dynamics (MD) simulation of miR193a-5p and APOL1 3′ UTR suggested a common root mean square deviation (RMSD) of 1.2 nm that seems to be converging but the radius of gyration (Rg) values show significant deviation during the 6–8 ns of simulation. Also, the root mean square fluctuation (RMSF) values of APOL1 3’ UTR shows a deviation in the base range 100–120 after 10 ns of simulation, while the miR193a show minimal deviation after 10 ns simulation. The miR193a base number 7–12, which belongs to the hairpin loop, show a little fluctuation during 5 ns and 10 ns of simulation. However, the seed region shows consistent trajectories after 5 and 10 ns simulations (Appendix A).

### 3.15. Evaluation of the Effect of VDA (Vitamin D Receptor Agonist) on Downing of miR193a and Associated Expression of Podocyte Molecular Markers

We asked if VDA could down-regulate miR193a in undifferentiated podocytes, would it be associated with enhanced mRNA expression of podocyte molecular markers? Both UPDs and DPDs were treated with VDA. After 48 h, RNAs were assayed for miR193a expression. Cumulative data of three independent experiments are shown in bar graphs (Figure 8A). VDA down-regulated the expression of miR193a both in UPDs (cultured at 33 °C) and DPDs (cultured at 37 °C). cDNAs from VDA- treated UPDs were also amplified with specific primers for *WT1*, *VDR*, *APOL1*, *CD2AP*, and *Nephrin*. VDA enhanced (*p* < 0.05) mRNA expression of podocyte markers (Figure 8B).

In brief, YY1/Sox2 expression as well down-regulation of WT1/ VDR: RXR de-represses the expression of miR193a supporting undifferentiated podocyte (UPDs) phenotype (Figure 9A); on the other hand, silencing of YY1/Sox2 as well as upregulation of WT1/VDR down-regulates the expression of miR193a promoting differentiated podocyte (DPD) phenotype (Figure 9B). Since VDR agonists achieves these outcomes through downing of miR193a, they could be a potential candidate for the enhancement of APOL1 expression and preservation of the podocyte molecular phenotype.

## 4. Discussion

Undifferentiated podocytes (UPDs) showed enhanced expression of miR193a, Sox2, and YY1, but down-regulation of APOL1, WT1, VDR, and RXR. On the other hand, differentiated podocytes (DPDs) displayed an opposite pattern- an attenuated expression of miR193a, YY1, and Sox2 but escalated expression of APOL1, WT1, and VDR. Interestingly, inhibition of miR193a in UPDs, escalated the expression of podocyte molecular markers, whereas overexpression of miR193a in DPDs resulted in the downing of APOL1 as well as podocyte molecular markers indicating a causal relationship between miR193a and podocyte molecular markers. Since silencing of YY1/Sox2 down-regulated the expression of miR193ain UPDs, suggested a causal relationship. On the other hand, the silencing of WT1, VDR, and RXR enhanced the expression of miR193a in DPDs, and it suggested their role as repressors. DPD lysates showed an increased expression of H3K27me^3^, and WT1-IP fraction revealed the presence of EZH2 and HDAC1, indicating that the WT1 repressor complex could be downing miR193a through methylation of lysine 27 residues at Histone (H) 3. Similarly, the RXR-IP fraction of DPD lysates showed the presence of VDR, SMRT, and HDAC3, suggesting that VDR-RXR heterodimers are bound with SMRT (a known nuclear co-repressor) and act as a repressor of miR193a. As expected, the treatment of undifferentiated podocytes with a VDR agonist (VDA) not only down-regulated the expression of miR193a but also enhanced the transcription of APOL1 as well as molecular markers of podocyte differentiation.

The MEME suite for motif identification suggested that transcription factor Sox2 can also bind to the miR193a promoter [20]. We generated miR193a promoter and transcription factor complexes using docking approaches and further studied the DNA-protein interactions [22,28,29,30,31,32,33]. The Sox2-miR193a gene interaction is the strongest in comparison to YY1 and WT1 in terms of free energy gain upon assembly formation and free energy of dissociation. The positive value of free energy of dissociation indicates that an external driving force should be applied to dissociate the assembly. The residue propensity of the DNA-protein interaction interface suggested that some basic amino acid residues and polar charged amino acids have an increased affinity for DNA interaction.

The MD simulations of miR193a promoter and various transcription factors including YY1, WT1, Sox2 and the nuclear receptors VDR-RXR heterodimer suggested that structures of these transcription factors are most stable during the simulation when they are bound with the DNA, however, there are some fluctuations in terms of root mean square fluctuation (RMSF) values of interacting residues, which may be minimized upon bond formations which is consistent with the thermodynamics analyses of such miR193a promoter and transcription factor complexes. The nuclear receptor VDR-RXR heterodimer binding with miR193a promoter shows clearly that the DNA binding domain of RXR plays a vital role in the interactions of VDR-RXR heterodimer with microRNA promoter. However, the substrate-binding domain of RXR seems to be also responsible for interaction with the miR193a promoter. Therefore, such interactions could be responsible for the regulation of miR193a expression in the cellular milieu as well.

There are several bioinformatics approaches to predict the association of microRNA (miRNAs) and mRNAs using sequence-based and structure-based methods to identify exact matches [2]. In the present study, we identified miR193a and APOL1 interaction based on the database miRbase [4], miRmap [25], and TargetScan [5,6] which provided the available binding site on the most prevalent and less prevalent transcripts of APOL1 mRNA. These features include the positioning of the miRNA binding site outside the path of the ribosome (which contains the first 15 nt of the 3’ UTR) and the positioning of the site within 3’ UTR segments that are more accessible to the silencing complex, as measured by high local AU content, high AU content of the entire 3’ UTR, shorter distance from a 3’ UTR terminus, shorter 3’ UTR length or less stable predicted competing secondary structure [5,6]. Moreover, two features of miRNA, including lower target site abundance (TA) within the transcriptome and stronger predicted seed pairing stability (SPS), can also influence site efficacy. We also got information about the binding and hybridization energies. This information indicates the complementary binding site of miR193a-5p to the 3’ UTR of APOL1, however, to understand the spatial features of miR193a and APOL1 3’ UTR interactions, we derived the secondary structures of APOL1 3’ UTR and miR193a. The secondary structures of APOL1 3’ UTR and miR193a suggested different components, including stem, loops, and hairpin structures that which conformation of heteroduplex is more stable in terms of energy [26,27]. Moreover, RNAs are in their native 3-dimensional conformation in the cellular milieu; therefore, the spatial features, including conformation and accessibility of the target site to the microRNA, seem to be important parameters to study. We generated the tertiary structures of APOL1 3’ UTR region and the miR193a and studied the miR193a-APOL1 3’ UTR interactions [28,29,30,31,32,33]. The Molecular Dynamics (MD) simulations of miR193a and APOL1 3’ UTR complex for 5 ns and 10 ns simulation suggested that the seed region of miR193a shows very consistent trajectories during the course of the simulation, while MD simulations of microRNA-mRNA complexes show more extended period for equilibration steps in case of TIP4P water and Amber99sb force field.

The WT1, Sox2, and YY1 alter the expression of genes involved in differentiation in general and during embryogenesis in particular [14,15]. Since silencing of either YY1 or Sox2 down-regulated the expression of miR193a in podocytes, it indicated a causal relationship between these molecules. The role of WT1 as a master transcriptor of podocyte markers is well recognized [42]. On that account, its expression plays a vital role in podocyte health. Notably, YY1, Sox2, and WT1 are involved in the differentiation of cells; therefore, the modulation of miR193a expression through them could be used as a therapeutic strategy. However, their broader effects on other cell types in different organs could be associated with side effects. On the other hand, Vitamin D is an indigenous ligand and being used therapeutically in a variety of disorders.

Vitamin D3 and VDR agonists (VDA) enhance VDR by two mechanisms, its bindings with VDR prevents proteasomal degradation of VDR as well as it enhances VDR transcription [43,44]. VDA bound VDR heterodimerizes with RXR and moves into the nucleus. VDA bound heterodimers may bind either on VDRE (vitamin D response element) or RXRE sites of the specific genes. Whether the VDR-RXR complex acts as an activator or repressor depends on the composition of the complex; if VDR-RXR recruits co-activators such as p300 CBP (CREB-binding protein) and HAT (Histone Acetyl Transferase), the complex acts as an activator; on the other hand, if it recruits nuclear co-repressors such as SMRT and HDAC3, the complex acts as a repressor. Interestingly, miR193a did not have any VDRE (Vitamin D Response Element) site but did have an RXRE (Retinoid X Response Element) site. Because RXR-IP fractions of DPDs showed the presence of not only VDR but also of SMRT and HDAC3, it indicated the formation of the repressor complex. Since the silencing of either VDR or RXR de-repressed the expression of miR193a in DPDs, it established the participation of both the molecules in the downing of miR193a in DPDs.

MiR193a has been shown to regulate the expression of WT1 in podocytes [42]. Interestingly, the recent study indicates that WT carries the potential to inversely restrict the expression of miR193a. Moreover, the WT1-IP fraction of DPDs showed the presence of SMRT, EZH2, and HDAC1; additionally, DPDs showed methylation at lysine 27 residues of Histone 3 (H3K27me3). Thus, it appears that WT1 is forming a repressor complex at miR193a. Since the silencing of WT1 de-repressed the expression of miR193a in DPDs, it further confirmed the functioning of the WT1 repressor complex in DPDs.

Down-regulation of miR193a has been reported in cells with enhanced expression of APOL1; the latter is also considered a marker of podocyte maturation [7,10,11,13]. On that account, VDA-induced down-regulation of miR193a could be used to sustain the podocyte molecular phenotype. VDA down-regulated miR193a both undifferentiated as well as differentiated podocytes. In reported studies, VDA treatment not only induced APOL1 in parietal epithelial cells but promoted the transition of parietal epithelial cells to podocytes [7]. The present study further confirms that modulation of miR193a can be used as a strategy to maintain the podocyte molecular phenotype.

We conclude the YY1 and Sox2 enhance, but WT1 and VDR-RXR repress the expression of miR193a in podocytes. miR193a expression inversely regulated the expression of podocyte markers. VDA-induced downing of miR193a carries the potential to enhance the expression of podocyte molecular markers.

## Figures and Tables

**Figure 1 cells-09-01004-f001:**
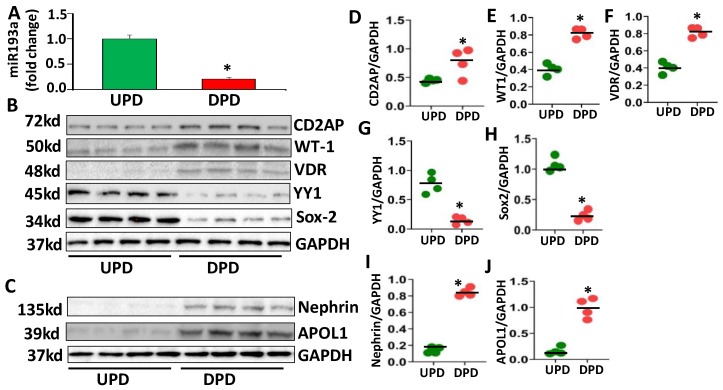
Podocyte molecular profiles in undifferentiated (UPD) and differentiated conditions (DPD). Podocytes were incubated in Petri dishes in media either at 33 °C for 48 h (UPD) or at 37 °C for 10 days (DPDs) (n = 4). (**A**) RNAs were extracted and assayed for miR193a (n = 4). Cumulative data are displayed in bar graphs (means ± SD). * *p* < 0.05 compared with UPD. (**B**) Protein blots from four independent lysates were probed for CD2AP, WT1, VDR, YY1, Sox2, and Glyceraldehyde -3-phosphate dehydrogenase (GAPDH). (**C**) Protein blots from four different lysates were probed for Nephrin, APOL1, and GAPDH. (**D**–**J**) Cumulative densitometric data (protein: GAPDH ratio) are shown in dot plots. * *p* < 0.05 compared with respective UPD.

**Figure 2 cells-09-01004-f002:**
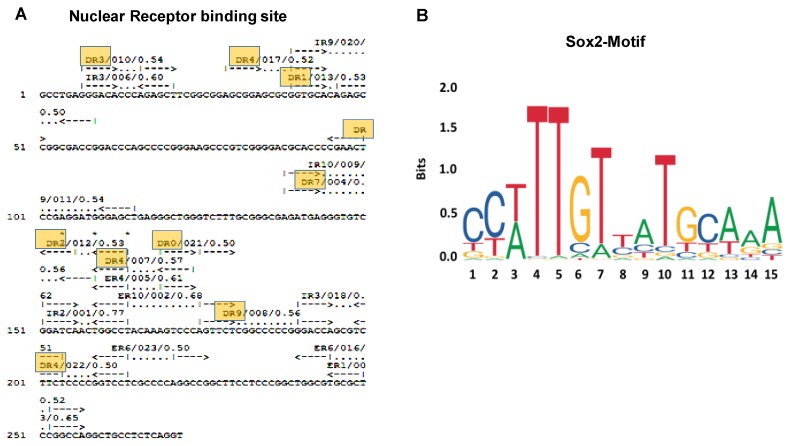
Transcription factor binding on miR193a promoter. (**A**). DR1, DR3 and DR4 elements for nuclear receptor binding. (**B**). Sox2 motif on the miR193a gene.

**Figure 3 cells-09-01004-f003:**
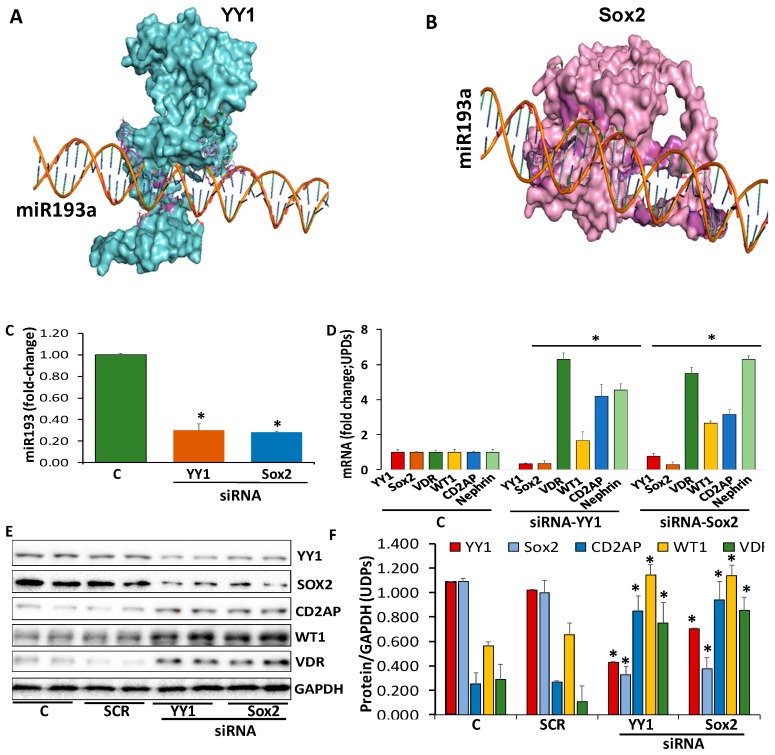
Evaluation of the transcription factors enhancing miR193a expression and associated downstream signaling. (**A**) YY1-miR193a complex. YY1 (Cyan) binds on miR193a promoter; the interacting residues (Magenta) are displayed. (**B**) Sox2-miR193a complex. Sox2 (Pink) binds on miR193a promoter and interacting residues (Magenta) are shown. (**C**) UPDs were transfected with siRNA-YY1 or SiRNA-Sox2 (n = 3). After 48 h, RNAs were extracted from control (C) and siRNA-YY-1 and siRNA-Sox2-transfected UPDs. RNAs were assayed for miR193a. Cumulative data are shown in a bar diagram. * *p* < 0.05 compared with control (C). (**D**) cDNAs were prepared from the RNAs extracted from the protocol A. cDNAs were amplified with specific primers for YY1, Sox2, VDR, WT1, CD2AP, and Nephrin (n = 3). Cumulative data are shown in a bar diagram. * *p* < 0.05 compared with respective controls (C). (**E**) UPDs were transfected with either scrambled (SCR), siRNA-YY1, or siRNA-Sox2 (n = 3). After 48 h, proteins were extracted from control, SCR-, siRNA-YY1- and siRNA-Sox2-transfected podocytes. Protein blots were probed for YY1, Sox2, CD2AP, WT1, VDR, and GAPDH. Representative gels from two different lysates are displayed. (**F**) Cumulative densitometric data from protein blots of protocol C are displayed in a bar diagram. * *p* < 0.05 compared with respective control and SCR.

**Figure 4 cells-09-01004-f004:**
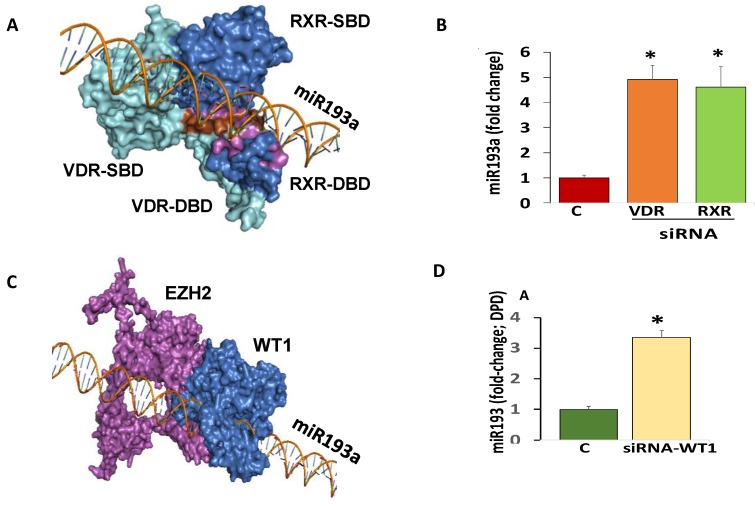
Determination of repressors contributing to the downing of miR193a expression. (**A**) Vitamin D receptor- Retinoid X receptor (VDR-RXR) heterodimer-miR193a complex. DNA binding domain (DBD) of RXR (Marine blue) and substrate-binding domain (SBD) of RXR (Marine blue) are involved in binding with DNA (miR193a). Some interacting residues of RXR-DBD and RXR-SBD are represented in Magenta color, and VDR-DBD and VDR-SBD are represented in Brown color. VDR substrate-binding domain (SBD) and DNA binding domain (DBD) is represented in Cyan color. (**B**) DPDs were transfected with either siRNA-VDR or siRNA-RXR (n = 3). After 48 h, RNAs were extracted from control (C) and siRNA-VDR- and siRNA-RXR-transfected DPDs. RNAs were assayed for miR193a. Cumulative data from three independent experiments are shown in bar graphs. * *p* < 0.05 compared with control (C). (**C**). WT1 recruits EZH2 and binds at miR193a promoter. WT1 (Marine blue) binds on miR193a promoter with interacting residues (Magenta). (**D**). DPDs were treated with buffer or transfected with siRNA-WT1 (n = 3). After 48 h, RNAs were extracted from control (C) and siRNA-WT1-transfected DPDs. RNAs were assayed for miR193a. Cumulative data from three independent experiments are shown in a bar diagram. * *p* < 0.05 compared with control (C).

**Figure 5 cells-09-01004-f005:**
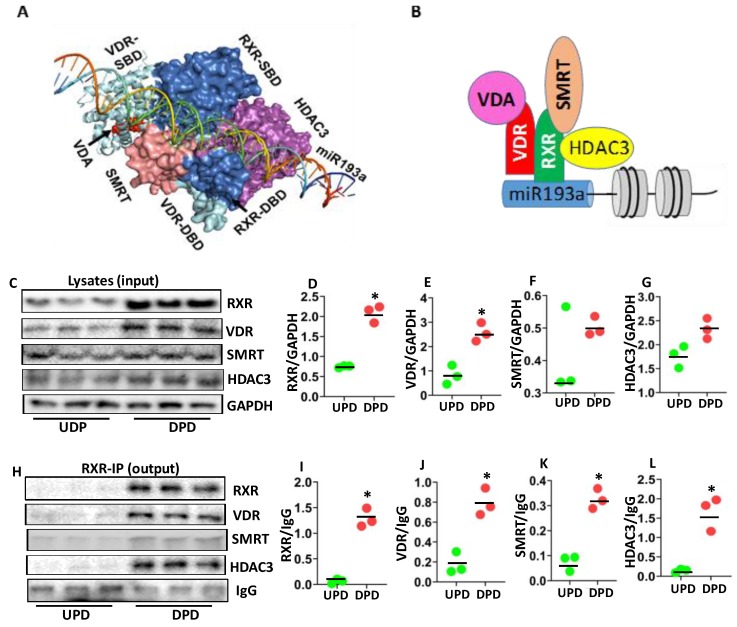
Analysis of VDR-RXR repressor complex. (**A**) The structural construct of the VDR-RXR repressor complex. The complex shows VDR (cyan) and RXR (marine blue) heterodimer. VDA (red) bound with VDR and RXR heterodimer interacts with miR193a promoter and recruits SMRT (pink) and HDAC3 (magenta) and forms a repressor complex. (**B**) A schematic diagram is displaying the formation of the VDR-RXR repressor complex at the miR193a promoter. The VDA bound VDR makes a heterodimer with RXR and recruits SMRT and HDAC3 and forms a repressor complex. (**C**) Protein blots of three different cellular lysates of UPDs and DPDs were probed for RXR, VDR, SMRT, HDAC3, and GAPDH. Gels of three independent lysates are displayed. (**D**–**G**). Cumulative densitometric data (protein GAPDH) are shown as dot plots. * *p* < 0.05 compared with respective UPD. (**H**) Cellular lysates from protocol C were immunoprecipitated (IP) with the RXR antibody. Protein blots of RXR-IP fractions were probed for RXR, VDR, SMRT, HDAC3, and IgG. Gels are displayed. (**I**–**L**). Cumulative densitometric data (protein/GAPDH) from the protocol H, are shown as dot plots. * *p* < 0.05 compared with respective UPD.

**Figure 6 cells-09-01004-f006:**
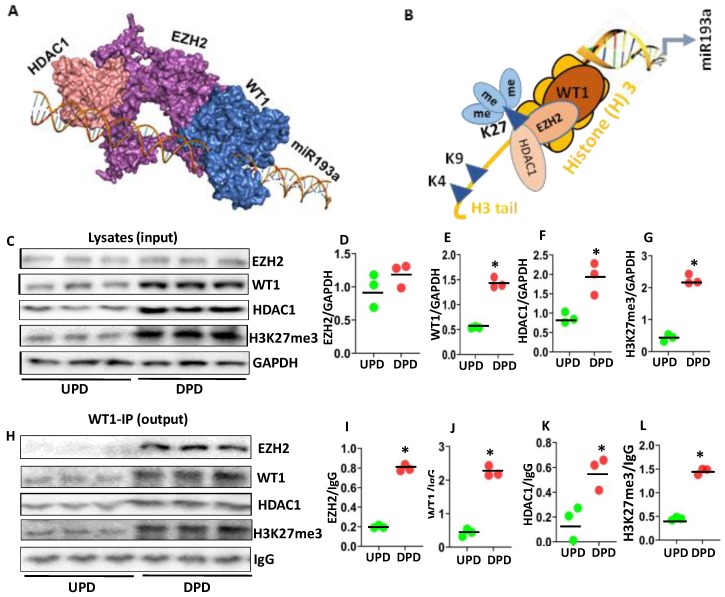
Analysis of the WT1-EZH2 repressor complex. (**A**) The structural construct of the WT1-EZH2 repressor complex. (**B**) A schematic diagram is displaying the formation of the WT1 repressor complex at the miR193a promoter. WT1 recruits EZH2, SMRT, and HDAC1 inducing methylation at lysine 27 residues at Histone (H) 3 tail. (**C**) Protein blots of three different cellular lysates of UPDs and DPDs were probed for WT1, EZH2, HDAC1, and H3K27me3, and GAPDH. Gels of three independent lysates are shown. (**D**–**G**). Cumulative densitometric data (protein/GAPDH) are shown as dot plots. * *p* < 0.05 compared with respective UPD. (**H**) Cellular lysates from protocol C were immunoprecipitated (IP) with the WT1 antibody. Protein blots of WT1-IP fractions were probed for WT1, EZH2, HDAC1, and H3K27me3, and IgG. Gels are displayed. (**I**–**L**) Cumulative densitometric data (protein/GAPDH) from the protocol H, are shown as dot plots. * *p* < 0.05 compared with respective UPD.

**Figure 7 cells-09-01004-f007:**
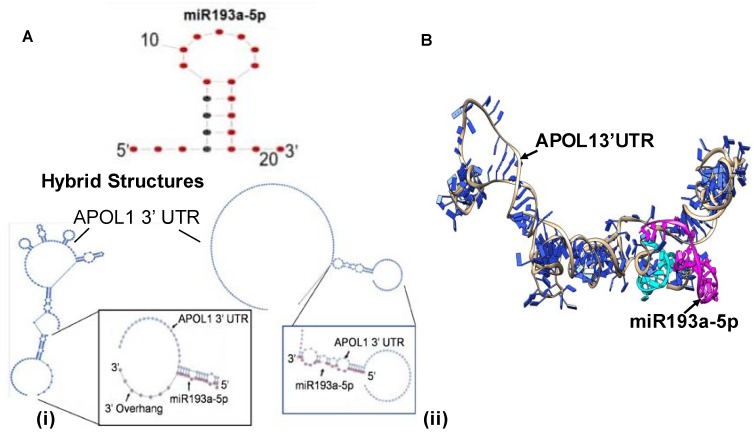
APOL1 3′ UTR and miR193a Secondary structures. (**A**) miR193a-5p structure conformation and hybrid structure conformations of APOL1 3′ UTR and miR193a-5p. The APOL1 3’ UTR structure conformation consists of secondary structure elements such as External loop, Helices, bulge loop, Interior loop, and Hairpin loops. i. The hybrid structure is showing miR193a-5p and APOL1 3′ UTR pairing with a 3′ overhang. ii. The hybrid structure conformation shows intensive pairing. (**B**) Tertiary structure of APOL1 3′ UTR and miR193a-5p (Magenta) complex. The miR193a-5p binding site on APOL1 3′ UTR is represented in Cyan. (**C**) UPDs were transfected with either control plasmid (empty vector, EV) or a specific inhibitor of miR193a (25 nM) (n = 3). In parallel sets of experiments, DPDs were transfected with either control plasmid (EV) or miR193a plasmid (50 nm) (n = 3). RNAs were assayed for miR193a. Cumulative data are displayed in bar graphs. * *p* < 0.0 compared to respective EV. (**D**) cDNAs from EV- and miR193a inhibitor-treated UPDs from 7C were amplified with specific primers for *APOL1*, *VDR*, *WT1*, and *Nephrin* (n = 3). Cumulative data are shown in a bar diagram. * *p* < 0.05 compared to respective EV. (**E**) cDNAs from EV and miR193a-transfected DPDs from 7C were amplified with specific primers for *APOL1*, *VDR*, *WT1*, and *Nephrin* (n = 3). Cumulative data are shown in bar graphs. * *p* < 0.05 compared with respective EV.

**Figure 8 cells-09-01004-f008:**
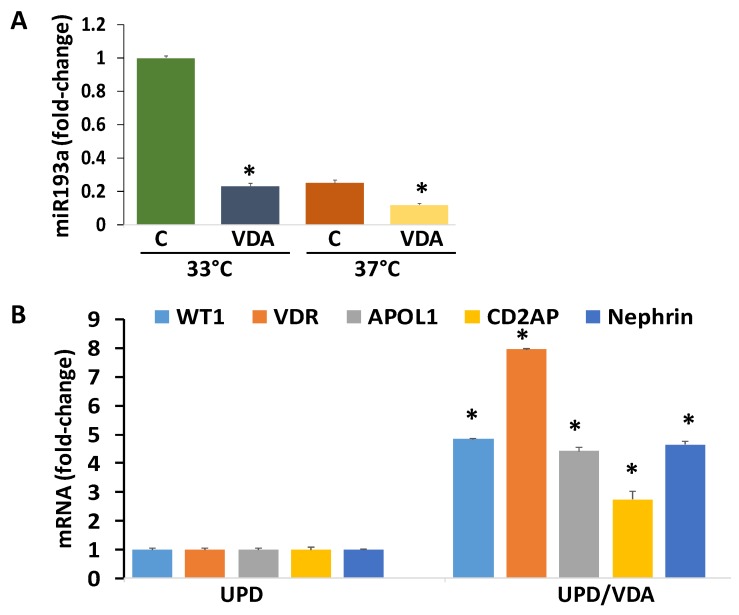
Evaluation of the effect of VDA on downing of miR193a and associated expression of podocyte molecular markers. (**A**) UPDs (at 33 °C) and DPDs (at 37 °C) were incubated in media containing either buffer (Control, C) or VDA (EB1089, 100 nM) for 48 h (n = 3). RNAs were assayed for miR193a expression. Cumulative data of three independent experiments are displayed in a bar diagram. * *p* < 0.05 compared with respective control (C). (**B**) B. cDNAs were prepared from RNA extracted from the protocol A, amplified with specific primers for *WT1*, *VDR*, *APOL1*, *CD2AP*, and *Nephrin*. Cumulative data are shown as bar graphs. * *p* < 0.05 compared respective controls (C).

**Figure 9 cells-09-01004-f009:**
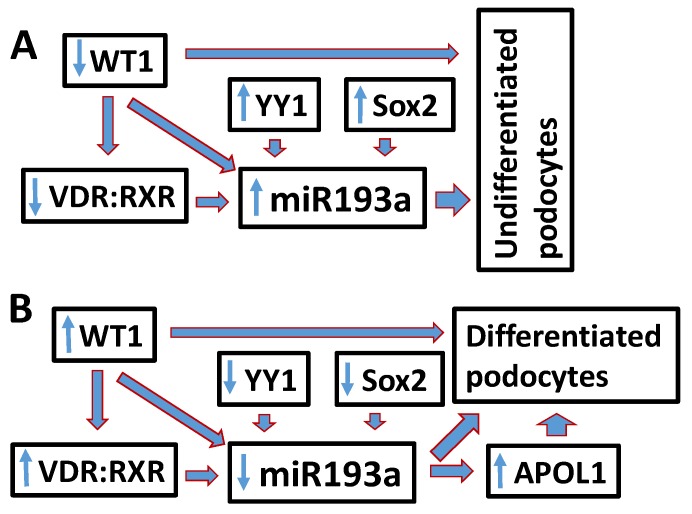
Schematic diagram showing modulation of miR193a expression determining the phenotype of podocytes. (**A**) Increased YY1/Sox2 expression enhances but own regulation of WT1 and VDR de-represses the expression of miR193a in UPDs. (**B**) Downing of YY1/Sox2 and escalation of WT1/VDR expression downregulates the expression of miR193a and maintains podocytes in a differentiated state.

**Table 1 cells-09-01004-t001:** DNA-Transcription factor interaction.

Complex	Surface Area (Å^2^)	ΔG^int^ (kcal/mol)	No. of Hydrogen bonds (N_HB_)	No. of Salt Bridges (N_SB_)	ΔG^diss^ (kcal/mol)	TΔS^diss^ (kcal/mol)
miR193a-YY1	2745.9	−50.4	160	0	107.3	14.2
miR193a-WT1	2812.8	−52.2	161	0	109.5	14.2
miR193a-Sox2	3165.8	−58.1	161	0	115.4	14.2
miR193a-VDR-RXR	11576.9	−76.4	161	3	5.8	12.6

**Table 2 cells-09-01004-t002:** Interacting residues in the DNA-Transcription factor complexes.

Complexes	Interacting Residues
miR193a-YY1	Glu19, Ile20, Pro32, Val33, Glu37, Thr38, Val40, Val41, Glu46, Asp49, Lys259, Asp269, Leu270, Ser271, Lys301, Gly302, Cys303, Thr304, Lys305, Glu336, Ser338, Lys339, Arg342, Leu366, Asp367, Phe368, and Lys401
miR193a-Sox2	Lys97, Arg100, Met104, Tyr110, Asn166, Ala194, Gln195, Met196, Gln197, Ser230, Thr234, Pro235, Gly236, Met237, Ser252, Ser253, Pro254, Pro255, Val256, and Arg264
miR193a-WT1	Ser315, Glu316, Arg321, Lys322, Tyr334, Phe335, His339, Lys346, Phe383, Gln384, Cys385, Lys386, Leu398, Lys399, Thr402, Met442, Thr443, and Lys444
miR193a-RXR(DBD) - miR193a-VDR-RXR heterodimer	Cys135, Ala136, Ile137, Leu167, Thr168, Tyr169, Thr170, Lys175, Asp176, Leu178, Ile179, Tyr189, Tyr192, Gln193, Lys194, Leu196, Ala197, and Met198.
miR193a-RXR(SBD) - miR193a-VDR-RXR heterodimer	Lys289, Ser295, Glu296, Leu297, Pro298, Leu299, Asp300, Gln302, Gly387, Ser389, Glu393, Met459, Leu460, Glu461, and Ala462.
miR193a-VDR(DBD) - miR193a-VDR-RXR heterodimer	Arg18, Asn19, Val20, Pro21, Arg22, Ile23, Cys24, Gly25, Cys27, Gly28, Asp29, Arg30, Met39, Arg67, Ile68, Thr69, Lys70, Asp71, Asn72, and Arg83.
miR193a-VDR(SBD) - miR193a-VDR-RXR heterodimer	Thr146, Gly289, Asn290, and Gln291

**Table 3 cells-09-01004-t003:** Comparison of miR193a-5p targets.

miRNA	Gene	ΔG Open	ΔG Total	ΔG Duplex	ΔG Binding	ΔG Duplex Seed	ΔG Binding Seed	3′UTR Position	AU Content	3′ Pairing
hsa-mir-193a-5p	APOL1	20.37	3.57	−16.80	−17.90	−8.70	−9.11	456.00	0.42	2.00
hsa-mir-193a-5p	BLID	19.96	5.16	−14.80	−15.97	−9.70	−10.17	92.00	0.79	1.00
hsa-mir-193a-5p	GPR17	19.42	7.33	−16.60	−17.70	−8.70	−9.11	125.00	0.34	2.00
hsa-mir-193a-5p	SYNPO2L	17.92	5.02	−12.90	−13.95	−9.70	−10.70	894.00	0.74	1.00
hsa-mir-193a-5p	NKIRAS1	15.35	0.15	−15.20	−16.23	−8.70	−9.11	641.00	0.51	2.00
hsa-mir-193a-5p	EIF2S1	12.90	−1.50	−14.40	−15.06	−9.70	−10.17	467.00	0.83	1.50

**Table 4 cells-09-01004-t004:** APOL1 3′ UTR-miR193a-5p interaction.

Complex	Surface Area (Å^2^)	ΔG^int^ (kcal/mol)	No. of Hydrogen bonds (N_HB_)	No. of Salt Bridges (N_SB_)	ΔG^diss^ (kcal/mol)	TΔS^diss^ (kcal/mol)
APOL1 3′ UTR- miR193a-5p	1341.3	−27.3	32	0	29.7	11.8

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
