# Peer review of "MiR193a Modulation and Podocyte Phenotype†"

_cells, 2020, doi:10.3390/cells9041004_

Round 1

Reviewer 1 Report

The revised manuscript is majorly enhanced in comparison to the previous version; this is mainly due to addition of (wet lab/in vitro) experimental results that give a reason to perform the in silico predictions.

In the text, three tables are mentioned (lines 352, 645, 721) which should summarize calculated values; these tables are not included/presented in the manuscript.

Line 357f: What is the difference between ΔGint and ΔGdiss?
Authors state in their rebuttal letter that ΔGint is the gain in Free energy upon complex assembly while ΔGdiss is the loss in Free energy upon complex dissociation; if this would be true, the values of both energies would be identical (except their sign), but here the values are quite different (ΔGint = -50.4 kcal/mol; ΔGdiss = 107.3 kcal/mol). That is, authors have to explain in the manuscript what these energies are.

Line 670ff: Any (m)RNA has a secondary structure consisting of an external loop, helices and bulge, internal and hairpin loops; thus delete the sentence "The APOL1 3' UTR structure conformation consists of secondary structure elements such as External loop, Helices, Bulge loop, Interior loop, and Hairpin loops (Fig. 11 A)"

Delete the sentence "The external loop (ΔG is -6.60 kcal/mol) has 35 single strand bases and 5 closing helices. There are 18 helices, 10 interior loops, 3 Bulge loops, and 5 Hairpin loops" because this gives only uninteresting things; such things could be added to one of the missing tables.

Fig. 11Ai, ii and C: Where are located these two different miRNA193a-5p binding sites shown in i) and ii) in the APOL1 3' UTR? What is the relevance of the binding site shown in C)? The latter one shows only one helix of six stacked base pairs while the complexes in i) and ii) clearly possess more base pairs.

Typos:
Line 70: precursor miRNA (pre-miRNA) => pre-miRNA
Line 78: expression APOL1 => expression of APOL1
Line 83: In vitro studies, => In in vitro studies,
Line 126: 10%fetal => 10% fetal
Line 165: from mRNA, => from RNA,
Line 176: U6-snuRNA => U6-snRNA
Line 178: (10-12) => [10-12]
Line 183: and transferred onto PVDF membranes were processed =>
and, after gel electrophoresis, transferred onto PVDF membranes
Line 298: expression of the profile => expression profile
Line 356: Lys401. (Fig. 3A) => Lys401 (Fig. 3A).
Line 427: DNA. (Fig. 4D) => DNA (Fig. 4D).
Line 447: (0.5312). (Fig. 4G) => (0.5312) (Fig. 4G).
Line 448: promoter. (Fig. 4H) => promoter (Fig. 4H).
Line 637: database. [24] => database [24].
Line 657: Bulge => bulge
Hairpin => hairpin

Reviewer 2 Report

The authors attempted to show the importance of miR193a in the regulation of APOL1 expression and differentiation of podocytes. Firstly, the authors showed that miR193a expression is regulated by transcription factors YY1 and Sox2, and repressor VDR and RXR. Secondly, the author reported that miR139a binds to APOL1 3'UTR  via various programmes only. It is necessary to provide solid experimental data, such as whether knockdown or overexpression of miR193a can directly affect APOL1 expression and podocyte differentiation.

My other concerns includes:

  1. The title is very misleading. As mentioned above, this manuscript provided no data support that miR193a regulates APOL1 expression and differentiation of podocytes. The only point they can claim is the factors potentially regulate miR193a in podocytes. 
  2. What is the point to highlight "wild type (WT)" for APOL1? 
  3. Most of the figures such as Fig 3, 4, 5, 7, 9, 11 and 12 shall be deleted. Authors can just briefly mention what they found or predicted by various programmes in text.  As all of these "evidence" are just predictions, need to prove it by bench works.

Round 2

Reviewer 1 Report

Authors have modified the revised manuscript according to my previous report.

Reviewer 2 Report

Authors addressed my concerns.